# *KRAS, NRAS, BRAF, HER2* and MSI Status in a Large Consecutive Series of Colorectal Carcinomas

**DOI:** 10.3390/ijms24054868

**Published:** 2023-03-02

**Authors:** Aleksandr S. Martianov, Natalia V. Mitiushkina, Anastasia N. Ershova, Darya E. Martynenko, Mikhail G. Bubnov, Priscilla Amankwah, Grigory A. Yanus, Svetlana N. Aleksakhina, Vladislav I. Tiurin, Aigul R. Venina, Aleksandra A. Anuskina, Yuliy A. Gorgul, Anna D. Shestakova, Mikhail A. Maidin, Alexey M. Belyaev, Liliya S. Baboshkina, Aglaya G. Iyevleva, Evgeny N. Imyanitov

**Affiliations:** 1Department of Tumor Growth Biology, N.N. Petrov Institute of Oncology, 197758 St. Petersburg, Russia; 2Department of Medical Genetics, St.-Petersburg Pediatric Medical University, 194100 St. Petersburg, Russia

**Keywords:** *KRAS*, *NRAS*, *BRAF*, microsatellite instability, *HER2*, colorectal cancer

## Abstract

This study aimed to analyze clinical and regional factors influencing the distribution of actionable genetic alterations in a large consecutive series of colorectal carcinomas (CRCs). *KRAS*, *NRAS* and *BRAF* mutations, *HER2* amplification and overexpression, and microsatellite instability (MSI) were tested in 8355 CRC samples. *KRAS* mutations were detected in 4137/8355 (49.5%) CRCs, with 3913 belonging to 10 common substitutions affecting codons 12/13/61/146, 174 being represented by 21 rare hot-spot variants, and 35 located outside the “hot” codons. *KRAS* Q61K substitution, which leads to the aberrant splicing of the gene, was accompanied by the second function-rescuing mutation in all 19 tumors analyzed. *NRAS* mutations were detected in 389/8355 (4.7%) CRCs (379 hot-spot and 10 non-hot-spot substitutions). *BRAF* mutations were identified in 556/8355 (6.7%) CRCs (codon 600: 510; codons 594–596: 38; codons 597–602: 8). The frequency of HER2 activation and MSI was 99/8008 (1.2%) and 432/8355 (5.2%), respectively. Some of the above events demonstrated differences in distribution according to patients’ age and gender. In contrast to other genetic alterations, *BRAF* mutation frequencies were subject to geographic variation, with a relatively low incidence in areas with an apparently warmer climate (83/1726 (4.8%) in Southern Russia and North Caucasus vs. 473/6629 (7.1%) in other regions of Russia, *p* = 0.0007). The simultaneous presence of two drug targets, *BRAF* mutation and MSI, was observed in 117/8355 cases (1.4%). Combined alterations of two driver genes were detected in 28/8355 (0.3%) tumors (*KRAS*/*NRAS*: 8; *KRAS*/*BRAF*: 4; *KRAS*/*HER2*: 12; *NRAS*/*HER2*: 4). This study demonstrates that a substantial portion of *RAS* alterations is represented by atypical mutations, *KRAS* Q61K substitution is always accompanied by the second gene-rescuing mutation, *BRAF* mutation frequency is a subject to geographical variations, and a small fraction of CRCs has simultaneous alterations in more than one driver gene.

## 1. Introduction

Colorectal cancer (CRC) affects approximately 1.9 million people per year, thus holding the third position in cancer morbidity worldwide [1]. Molecular genetic testing has become an essential component of CRC management. Patients with metastatic CRC usually receive *KRAS*, *NRAS*, *BRAF*, microsatellite instability (MSI) and *HER2* testing [2,3]. *KRAS*/*NRAS* analysis is complicated because of a wide spectrum of activating mutations affecting these genes [4]. While tumors with wild-type *RAS* genes are highly sensitive to anti-EGFR therapy, erroneous administration of cetuximab or panitumumab to patients with *RAS*-mutated CRC may facilitate tumor growth [5,6]. CRCs carrying amino acid substitutions in codon 600 have a particularly poor prognosis and are potentially responsive to the combination of BRAF inhibitors and anti-EGFR antibodies [7]. *HER2*-driven CRCs can be managed by various antagonists of HER2 kinase [8]. Microsatellite instability occurs in CRCs caused by Lynch syndrome, as well as in a subset of sporadic cancers. The identification of MSI in CRC tissue may call for germline DNA testing. In addition, microsatellite unstable tumors are sensitive to immune checkpoint inhibitors. MSI analysis is employed not only for the management of metastatic tumors but also for patients with early-stage CRC [9,10].

The frequencies of genetic alterations observed in CRC patients vary across studies. It is very likely that technical bias contributes to these variations. *RAS* testing was initially limited to *KRAS* exon 2 and 3 hotspot mutations, and was supplemented by so-called “extended” *RAS* analysis (hotspot substitutions in *KRAS* exon 4 and *NRAS* exons 2, 3 and 4) only a few years ago [4,11]. Reliable analysis of *KRAS*/*NRAS* status still presents a challenge, as several available commercial assays limit the detection of *RAS* mutations to a number of relatively common events, thus missing a substantial portion of clinically relevant genetic alterations. Sanger sequencing and pyrosequencing are capable of detecting the entire spectrum of mutations. However, these sequencing methods are not efficient in tumor samples containing a low proportion of tumor cells. Next-generation sequencing (NGS) is certainly the method of choice. However, its use is still limited due to high cost and the need to accumulate multiple samples for a single run [4,6,11,12]. 

We have developed an inexpensive CRC diagnostic pipeline, which is capable of detecting both “typical” and “atypical” mutations in *KRAS*, *NRAS* and *BRAF* genes, even in samples with a low proportion of tumor cells, includes *HER2* and MSI testing, and is characterized by low cost. This uniform methodology was applied to 8355 consecutive CRC samples obtained from various parts of Russia. This study provides interesting insights into region-specific variations in mutation frequencies, ratios between “common” and “rare” genetic alterations, and the co-occurrence of multiple driver mutations.

## 2. Results

*KRAS* mutations were identified in 4137/8355 (49.5%) CRCs (Appendix A). A total of 3913/4137 (94.6%) mutations affected hot-spots and had a frequency above 1% among *KRAS*-mutated CRCs (G12D: 1193 (28.8%), G12V: 874 (21.1%), G13D: 727 (17.6%), G12C: 276 (6.7%), A146T: 235 (5.7%), G12A: 206 (5.0%), G12S: 200 (4.8%), Q61H: 108 (2.6%), A146V: 51 (1.2%), G12R: 43 (1.0%)). A total of 174/4137 (4.2%) CRCs carried 21 rare variants affecting the hot-spots; amino acid substitutions (Q61L: 35 (0.9%), G13C: 24 (0.6%), Q61R: 24 (0.6%), A59T: 21 (0.5%), Q61K: 21 (0.5%), G13R: 10 (0.2%), A146P: 6 (0.2%), A59E: 6 (0.2%), A59G: 6 (0.2%), G12F: 5 (0.1%), Q61P: 4 (0.1%), G13S: 2 (0.05%); G12L, G13V, Q61D: 1 (0.02%) each) or non-missense variants (G13dup: 2 (0.05%); A59del, G12Rfs*22, G12Sfs*22, G13_V14delinsDI, G60_Q61delinsE: 1 (0.02%) each) were present. A total of 35/4137 (0.9%) CRCs had mutations located outside the “hot” codons (V14I: 6 (0.2%), A18D: 5 (0.1%), L19F: 5 (0.1%), Q22K: 5 (0.1%), G60D: 4 (0.1%); A66X, E62K, E63del, G10_A11dup, G10dup, G10R, G10V, K147E, L19_T20delinsFS, T58I: 1 (0.02%) each), with most of them resulting in the activation of KRAS protein [4]. In addition, 12/4137 (0.3%) tumors carried two mutations, and one CRC had three distinct *KRAS* mutations. *KRAS* mutations were more common in females (52.0% vs. 47.0%, *p* < 0.0001) and in patients aged above 50 years (50.2% vs. 45.3%, *p* = 0.002) (Appendix A, Figure 1 and Figure 2). The distribution of *KRAS* mutation frequencies was relatively even across different geographic regions (Appendix A, Figure 3). 

A recent study revealed that *KRAS* Q61K mutation is not activating per se as it leads to the aberrant splicing and disruption of the gene function [13]. Instead, Kobayashi et al. [13] have shown that when occurring in tumors, *KRAS* Q61K is almost always accompanied by another mutation (usually a G60G silent mutation affecting codon 60), which restores the normal processing of the *KRAS* RNA transcript. *KRAS* Q61K substitution was observed in 21 CRCs in our data set, with 19 specimens available for pyrosequencing analysis. Strikingly, all 19 analyzed tumors indeed carried a second *KRAS* alteration, with GQ60_61GK function-restoring mutation being the most prevalent (17/19, 89.5%; c.180_181delinsAA: 15; c.180_181delinsCA: 2).

*NRAS* mutations were detected in 389/8355 (4.7%) CRCs. Several hot-spot mutations were relatively common (Q61K: 95 (24.4%), G12D: 67 (17.2%), Q61R: 59 (15.2%), Q61L: 38 (9.8%), G13R: 21 (5.4%), Q61H: 19 (4.9%), G12V: 18 (4.6%), G13D: 18 (4.6%), G12C: 14 (3.6%), G12S: 12 (3.1%), G12A: 7 (1.8%), G13V: 6 (1.5%)). There were 10/389 (2.6%) CRCs carrying missense mutations located outside hot spots (G60E: 2 (0.5%); A11T, A18T, A59T, A66V, E62K, E63D, G15E, Y64C: 1 (0.3%) each). *NRAS* mutations were slightly more prevalent in males (5.3% vs. 4.0%, *p* = 0.004). Their occurrence was not affected by the age at diagnosis or geographic region (Appendix A; Figure 1, Figure 2 and Figure 3).

*BRAF* mutations were identified in 556/8355 (6.7%) CRCs. Of these, 510/556 (91.7%) were kinase-activating codon 600 substitutions (509 V600E and 1 V600K). A total of 38/556 (6.8%) CRCs carried mutations affecting codons 594, 595 or 596, which result in the down-regulation of BRAF kinase activity but increased ERK signaling via bypass mechanisms [14,15]. There were also rare instances of *BRAF*-activating mutations affecting codons 597 (*n* = 2), 599 (*n* = 2), 601 (*n* = 3) and 602 (*n* = 1). *BRAF* mutations were almost twice more common in females than in males (8.4% vs. 4.9%, *p* < 0.0001) and tended to have different frequencies in different age groups (17–30 years old: 1/45 (2.2%), 31–40: 18/288 (6.3%), 41–50: 39/796 (4.9%), 51–60: 139/1925 (7.2%), 61–70: 217/3458 (6.3%), 71–80: 122/1613 (7.6%), >81: 19/217 (8.7%)) (Appendix A, Figure 1 and Figure 2). In contrast to *RAS* mutations, *BRAF* mutation frequencies were subject to geographic variation, with a relatively low occurrence in areas with an apparently warmer climate (4.8% in Southern Russia and North Caucasus vs. 7.1% in other regions of Russia, *p* = 0.0007) (Appendix A, Figure 3 and Figure 4). 

MSI was detected in 432/8355 (5.2%) CRCs. There was a striking increase in MSI frequency in patients below 40 years of age (48/333, 14.4%), which is almost certainly related to the high proportion of Lynch syndrome in younger individuals [2]. Patients aged 51–70 years demonstrated relatively low MSI occurrence (318/6996, 4.5%). A trend towards elevated MSI incidence in older age groups (71–80: 87/1613 (5.4%); >81: 16/217 (7.4%), *p* < 0.0001) was observed (Appendix A, Figure 2). The frequency of detection of MSI was 317/6324 (5.0%) when using a single marker BAT26, and 115/2031 (5.7%) with the pentaplex panel. Out of 115 CRCs with MSI detected by the pentaplex panel, only 4 did not show the instability for the BAT26 marker.

*BRAF* mutations were detected in 117/432 (27.1%) microsatellite-unstable CRCs. Double MSI/*BRAF*-positive tumors were significantly more common in women than in men (92/4151, 2.2% vs. 25/4204, 0.6%, *p* < 0.0001) and in elderly individuals (0.09% in patients below 50 years, 1.5% in 51–80 years age group and 3.7% in individuals older than 80 years, *p* < 0.0001).

Approximately one third of MSI-positive cases contained *KRAS* alterations (138/432, 31.9%). The simultaneous occurrence of MSI and *KRAS* mutations decreased with age (4.3% in patients below 50 years, 1.2% in 51–80 years age group and 0.9% in individuals older than 80 years, *p* < 0.0001).

HER2 activation by gene amplification and overexpression was detected in 99/8008 CRCs (1.2%) (Appendix A). No significant correlation with clinical characteristics of the disease was observed, although the statistical analysis was compromised by the low frequency of this event. 

A small fraction of CRCs contained combinations of mutations in distinct driver genes (*KRAS* and *NRAS* mutations: 8 (0.1%), *KRAS* and *BRAF* mutations: 4 (0.05%), *KRAS* mutation and *HER2* amplification/overexpression 12 (0.15%), *NRAS* mutation and *HER2* amplification/overexpression: 4 (0.05%)) (Appendix A). The total frequency of combined driver events was 26/8355 (0.3%).

## 3. Discussion

This study describes a large series of consecutive CRCs, which were collected in various regions of Russia and genetically tested using a low-cost, but nevertheless comprehensive, methodology capable of evaluating the entire spectrum of medically relevant alterations (hot-spot and rare mutations in *KRAS, NRAS* and *BRAF* oncogenes; *HER2* amplification/overexpression; MSI). Strikingly, the observed frequency of *RAS* mutations is on the upper limit of inter-study variations, suggesting that the combination of HRM, allele-specific PCR and pyrosequencing is reliable for detection of both “typical” and “atypical” mutations [11,16,17,18]. For example, the occurrence of *KRAS* alterations in the current study (49.5%) was higher than that reported at cBioPortal, a resource hosting genomic data from large cancer sequencing consortiums and individual studies (44.7%) (Appendix A) [19,20]. Our study underscores that clinical CRC *RAS* testing should not be limited to allele-specific PCR, as a substantial portion of *RAS* mutations is destined to be missed by this approach [4,6,21]. This is an important finding, given that the anti-EGFR therapeutic antibodies cetuximab and panitumumab are contraindicated for *RAS*-mutated CRCs and may even boost tumor progression in this category of patients [5,6].

*KRAS* Q61K is a rare genetic event. A recent study led to a surprising finding suggesting that Q61K substitution inactivates the *KRAS* gene if present alone, but is almost always accompanied by the second function-rescuing mutation in naturally occurring tumors [13]. The large size of our data set permitted us to analyze a substantial number of CRCs carrying *KRAS* Q61K substitution, and we provide, apparently, the first independent confirmation of the report of Kobayashi et al. [13].

There are hundreds of reports describing the distribution of *KRAS* and *NRAS* mutations in CRC patients. Although patients’ race, age, gender and other factors appear to have some impact, it seems that the differences in the observed frequencies are significantly more attributed to the variations in the methodology of the mutation testing than to genuine clinical or biological reasons [11,18,22]. Indeed, our study confirmed the mild influence of age and gender on the probability of detecting *RAS* activation [11,16,19,20] (Appendix A), while the pattern of *RAS* mutations was relatively uniform across various regions of Russia, suggesting a limited impact of lifestyle, environmental or other external factors. 

In contrast, this investigation revealed strong regional differences in the distribution of *BRAF* mutations, suggesting that this event is more characteristic of areas with a relatively cold climate. This is an interesting observation, with either diet, lifestyle or other climate-related factors assuming a role in determining the probability of developing *BRAF*-mutated CRC disease. Previous studies revealed race-specific differences in the distribution of *BRAF* mutations, i.e., an increased prevalence of this event in CRC patients of European vs. African or Asian descent [18,23]. Some investigations reported ethnic variations with regard to the frequency of *BRAF* alterations in CRC observed within the subjects of the same race [24]. In our study, a low rate of *BRAF* oncogene involvement was detected in patients from the North Caucasus and Southern Russia (Figure 4). While the population of the North Caucasus is represented by White non-Slavic people, the ethnic composition of the Southern Russia is identical to other regions of this country. There is also some evidence for a moderate contribution of smoking or dietary factors in determining the probability of development of *BRAF*-mutated CRC [25,26,27]. 

The comparison of data obtained for *HER2* activation and MSI with other studies is a complicated task. *HER2* overexpression is usually analyzed using immunohistochemical staining [28,29]. We have incorporated the determination of *HER2* status into the molecular genetic pipeline using the analysis of *HER2* extra copies as a primary test and the quantitation of the *HER2* RNA transcript as a confirmation of the functional relevance of *HER2* amplification. This approach, although promising, still needs to be rigorously validated against clinically accepted methodologies of *HER2* CRC testing. MSI frequencies are significantly influenced by several factors. MSI is a relatively common occurrence in the early-onset CRC; however, its incidence may depend on a population-specific incidence of Lynch-syndrome-associated germ-line pathogenic variants. Furthermore, MSI is particularly characteristic of very elderly subjects, so the age distribution of a given CRC patient series is a strong confounding factor [2].

The driver mutations affecting *KRAS*, *NRAS*, *BRAF* and *HER2* genes are generally mutually exclusive. This study along with similar reports and cBioPortal data describes rare instances of combined alterations of the above genes [19,20,30,31] (Appendix A). It is of question whether this phenomenon reflects intratumoral genetic heterogeneity, i.e., the situation where distinct cell clones carry distinct genetic alterations, or true instances of the co-occurrence of several driver events in the same cell. The combination of *HER2* amplification/overexpression with *RAS* mutations seems to be particularly common, being detected in 16/99 (16%) *HER2*-associated tumors. The responsiveness of these CRCs to HER2-targeted therapy needs to be evaluated in clinical studies.

In conclusion, this investigation produced several findings of potential importance. Atypical *KRAS* and *NRAS* mutations represent a substantial portion of *RAS* alterations which need to be considered in clinical testing. *KRAS* Q61K substitution is a gene-inactivating event if occurring alone, but it is always accompanied by the second function-rescuing mutation in naturally occurring CRCs. The frequency of *BRAF* but not other CRC-specific genetic aberrations may be a subject of climate-related variations. There are rare instances of CRCs carrying simultaneous alterations in several genes involved in MAPK signaling cascade. The analysis of biological mechanisms underlying the latter two observations deserves further consideration.

## 4. Materials and Methods

This study included 8355 consecutive CRCs, which were referred for molecular genetic analysis to the N.N. Petrov Institute of Oncology (St.-Petersburg, Russia) within years 2021–2022. Formalin-fixed paraffin-embedded (FFPE) CRC samples were subjected to microscope-guided manual tumor cell dissection, and nucleic acids (DNA and RNA) were extracted from the tumor cells using Trizol reagent as described previously [32]. In brief, tissue sections were washed with 70% ethanol, air-dried and then incubated overnight in 200 µL of lysis buffer (10 mM Tris–HCl (pH 8.0), 0.1 mM EDTA (pH 8.0), 2% SDS, 20 mg/mL proteinase K) at 65 °C. After sample cooling at room temperature, 200 μL Trizol and 90 μL chloroform–isoamyl alcohol mix (24:1) were added, samples were shaken rigorously and centrifuged at full speed (15,000× *g*) for 15 min at 0 °C. The supernatant was transferred into new tubes, to which 1 μL of glycogen (20 mg/mL) and 1 volume (300 μL) of cold isopropanol were added. The samples were vortexed and left overnight at −20 °C. The tubes were then centrifuged at 15,000× *g* for 30 min. Isopropanol was removed, and the precipitate was rinsed once in 70% ethanol for 10 min. After thorough removal of ethanol, the precipitate was dried at 50 °C, and then dissolved in 100 µL of sterile water at 50 °C for 5 min. RNA was enzymatically converted into cDNA only in samples positive for *HER2* gene amplification [33]. The reaction setup included two steps. First, 10 µL of nucleic acid sample was mixed with 1 µL dNTP mix (25 µmol each) and 2 µL of hexaprimers (0.25 µmol) in a total volume of 15 μL and incubated for 3 min at 70 °C, 3 min at 65 °C and 1 min at 60 °C in order to denature RNA and anneal primers. After sample cooling on ice, 4 µL 5^X^ RT buffer (Amgen, Thousand Oaks, CA, USA), 0.3 µL M-Mulv reverse transcriptase (Amgen, Thousand Oaks, CA, USA), 0.2 µL and RiboLock RNase Inhibitor (Thermo Scientific, Waltham, MA, USA) were added. The total volume of the reverse transcription (RT) reaction was 20 μL. Reaction conditions were 20 °C for 5 min, 38 °C for 30 min and 95 °C for 5 min. After the completion of cDNA synthesis, 80 μL of water was added to the sample. cDNA quality was checked by the Cycle threshold (Ct) of the housekeeping gene *SDHA* obtained in qPCR. Samples with *SDHA*(Ct) < 34 cycles were considered suitable for further analysis of *HER2* expression. 

Testing for *KRAS*, *NRAS* and *BRAF* mutations was performed by a combination of high-resolution melting (HRM) analysis, allele-specific PCR (AS-PCR), digital droplet PCR and pyrosequencing. First, the presence of *KRAS* (exons 2, 3, and 4), *NRAS* (exons 2, 3, and 4), and *BRAF* (exon 15) alterations was determined by HRM of PCR products. Cases showing abnormal melting patterns were further tested for hot-spot variants by the corresponding AS-PCR assays (*KRAS* exon 2: codons 12, 13; exon 3: codons 59, 61; exon 4: codon 146; *NRAS* exon 2: codons 12, 13; exon 3: codon 61; *BRAF* exon 15: codon 600). Tumor samples with equivocal results were additionally analyzed by digital droplet PCR. Cases with abnormal HRM curves negative for relevant hot-spot variants were subjected to pyrosequencing. The list of the primers, assay conditions and utilized equipment is given in Appendix A.

*HER2* gene amplification was determined by a quantitative real-time PCR assay (Appendix A). Tumor samples with extra copies of *HER2* DNA were subsequently tested for *HER2* mRNA overexpression. The thresholds for *HER2* DNA amplification and RNA overexpression were determined by comparing the IHC/FISH-validated HER2-positive and negative control samples. These thresholds were dCt < 0 and dCt < −1.9 for DNA amplification and RNA overexpression, respectively, where dCt = Ct*_HER2_* − Ct_reference_. 

Microsatellite instability (MSI) status was evaluated by fragment analysis of either a single marker (BAT26; 6324 samples) or five mononucleotide markers (BAT25, BAT26, NR21, NR22 and NR24; 2031 samples) using the GenomeLab GeXP Genetic Analysis System (Beckman Coulter, Brea, CA, USA) (Appendix A). For pentaplex panel, tumors with two or more shifts were classified as MSI-positive.

Mutation frequencies and their associations with clinical parameters were analyzed using a Chi-square test with Yates correction or Fisher’s exact test. Statistical comparisons were performed using R software (version 3.2.1, http://www.r-project.org (accessed on 16 February 16 2023)). The level of statistical significance was set at α = 0.05.

## Figures and Tables

**Figure 1 ijms-24-04868-f001:**
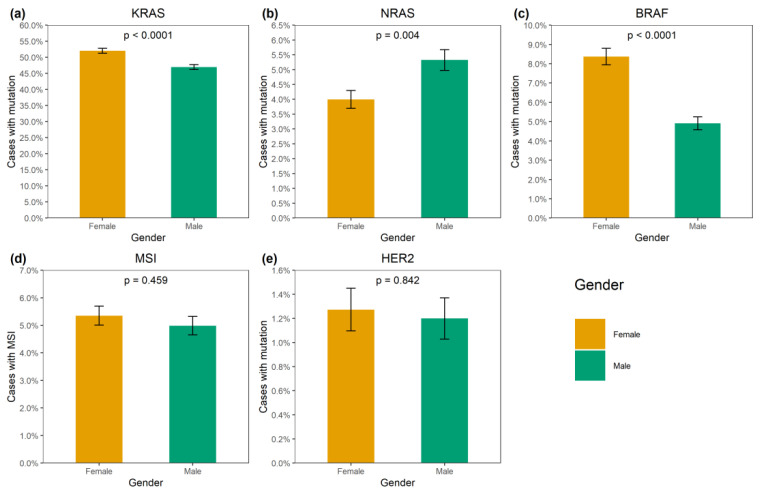
Association of genetic alterations with gender. (**a**) *KRAS* mutation, (**b**) *NRAS* mutation, (**c**) *BRAF* mutation, (**d**) MSI, (**e**) *HER2* amplification.

**Figure 2 ijms-24-04868-f002:**
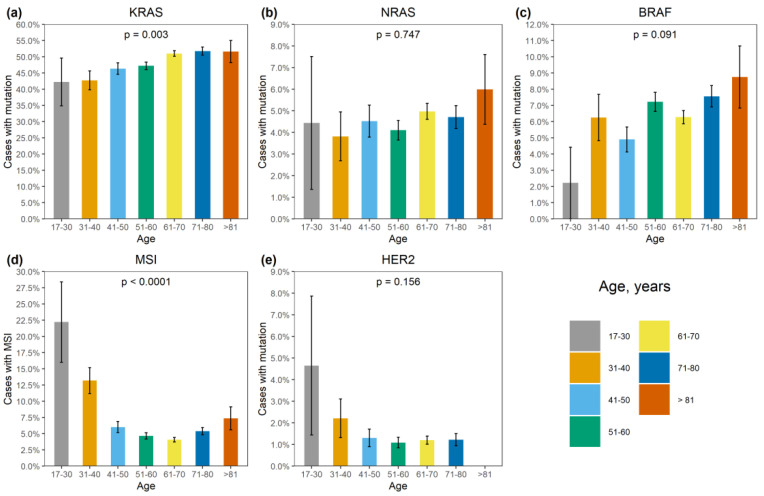
Association of genetic alterations with age. (**a**) *KRAS* mutation, (**b**) *NRAS* mutation, (**c**) *BRAF* mutation, (**d**) MSI, (**e**) *HER2* amplification.

**Figure 3 ijms-24-04868-f003:**
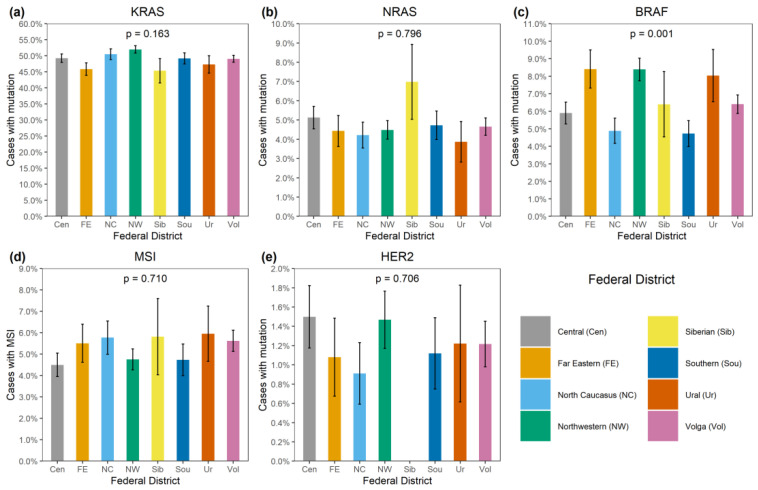
Distribution of genetic alterations in patients from various regions of Russia. (**a**) *KRAS* mutation, (**b**) *NRAS* mutation, (**c**) *BRAF* mutation, (**d**) MSI, (**e**) *HER2* amplification.

**Figure 4 ijms-24-04868-f004:**
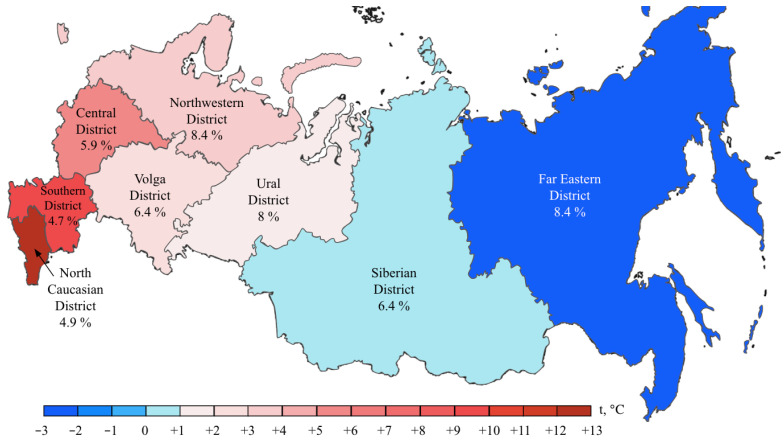
*BRAF* mutation frequencies in CRC patients from various region of Russia. The color bar represents an average annual temperature.

## Data Availability

The data that support the findings of this study are available from the corresponding author upon reasonable request.

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
