# Peer review of "KRAS, NRAS, BRAF, HER2 and MSI Status in a Large Consecutive Series of Colorectal Carcinomas"

_ijms, 2023, doi:10.3390/ijms24054868_

Round 1
Reviewer 1 Report
I read with interest the paper by Aleksandr S. Martianov et al titled "KRAS, NRAS, BRAF, HER2 and MSI status in a large "consecutive series of colorectal carcinomas". In my opinion, the strengths of this study are first of all the numerosity of the analyzed cohort consisting of 8355 colorectal cancer patients and the geographical location of these patients, distributed over a large region which is Russia.
However, there are some points that should be expanded and further investigated.
First, mainly it appears that there is some "insistence" on geographical variations and regional factors in the study without making any mention of ethnic factors. The very fine Figure 4 on the distribution of BRAF mutations shows markedly different frequencies between western and eastern districts. Is the cause to be found only in climatic factors? It seems strange that throughout the paper no reference is made to ethnic factors. On the other hand, data from the literature (such as DOI: 10.1158/1055-9965.EPI-08-0091; DOI: 10.1158/1055-9965.EPI-09-1112; DOI: 10.3892/ol.2015.3560; DOI: 10.1093/jnci/djv186) should be taken into account in this context. In addition, there is not even the slightest mention of different dietary factors, which could be an interesting source of discussion.
I noticed that the frequency of mutations is reported on the total number of mutations but not on the number of patients. For example, Q61K (present in 21 patients) is reported with a frequency of 0.5% considering 4137 KRAS-positive cases, but in fact the frequency over the total number of 8355 patients would actually be 0.25%, in perfect agreement on the other hand with that reported by cBioPortal.
I suggest also adding the frequency relative to the entire cohort of patients in Table S1. I made this premise because I think that, to make the study more interesting, it would be the case that the authors should compare the results with the data reported in cbioPortal (https://www.cbioportal.org/). It is my opinion that this large and so accurate study should be completed with a comparison of the results with those deposited in a dataset. In this way, the paper could be of greater impact, attraction and interest to readers.
Author Response
Comment: First, mainly it appears that there is some "insistence" on geographical variations and regional factors in the study without making any mention of ethnic factors. The very fine Figure 4 on the distribution of BRAF mutations shows markedly different frequencies between western and eastern districts. Is the cause to be found only in climatic factors? It seems strange that throughout the paper no reference is made to ethnic factors. On the other hand, data from the literature (such as DOI: 10.1158/1055-9965.EPI-08-0091; DOI: 10.1158/1055-9965.EPI-09-1112; DOI: 10.3892/ol.2015.3560; DOI: 10.1093/jnci/djv186) should be taken into account in this context. In addition, there is not even the slightest mention of different dietary factors, which could be an interesting source of discussion.
Response: Thank you for this very valuable comments. The discussion on the possible role of ethnicity, smoking and dietary factors has been incorporated in the paper:
…Previous studies revealed race-specific differences in the distribution of BRAF mutations, i.e., an increased prevalence of this event in CRC patients of European vs. African or Asian descent [18,23]. Some investigations reported ethnic variations with regard to the frequency of BRAF alterations in CRC observed within the subjects of the same race [24]. In our study, low rate of BRAF oncogene involvement was detected in patients from the North Caucasus and Southern Russia (Figure 4). While the population of the North Caucasus is represented by non-Slavic people of the White race, the ethnic composition of the Southern Russia is identical to other regions of this country. There is also some evidence for a moderate contribution of smoking or dietary factors in determining the probability of development of BRAF-mutated CRC [25-27].
Comment: I noticed that the frequency of mutations is reported on the total number of mutations but not on the number of patients. For example, Q61K (present in 21 patients) is reported with a frequency of 0.5% considering 4137 KRAS-positive cases, but in fact the frequency over the total number of 8355 patients would actually be 0.25%, in perfect agreement on the other hand with that reported by cBioPortal.
Response: A column reporting the frequency of each individual mutation in the entire patient cohort was added to the Table S1.
Comment: I suggest also adding the frequency relative to the entire cohort of patients in Table S1. I made this premise because I think that, to make the study more interesting, it would be the case that the authors should compare the results with the data reported in cbioPortal (https://www.cbioportal.org/). It is my opinion that this large and so accurate study should be completed with a comparison of the results with those deposited in a dataset. In this way, the paper could be of greater impact, attraction and interest to readers.
Response: Table S1 now contains the data on mutation frequencies in the entire patient cohort. Two additional Supplementary Tables, Table S5 and Table S6, comparing the data on KRAS, NRAS, BRAF mutations obtained in the current study and available at cBioPortal, were incorporated in the manuscript. The Discussion section now also refers to the cBioPortal database:
… For example, the occurrence of KRAS alterations in the current study (49.5%) was higher than that reported at cBioPortal, a resource hosting genomic data from large cancer sequencing consortiums and individual studies (44.7%) (Supplementary Table S5) [19-20]. …
… Indeed, our study confirmed a mild influence of the age and gender on the probability of detecting RAS activation [11,16, 19-20] (Supplementary Tables S2, S6). …
....This study along with similar reports and cBioPortal data describes rare instances of combined alterations of the above genes [19,20,30-31] (Supplementary Table S5).
Reviewer 2 Report
The English language and style are fine. The "low-cost but nevertheless comprehensive methodology" has not been described and shown. Materials and Methods section is brief. The conclusion about "BRAF mutation frequency is a subject to geographical variations, with a relatively low occurrence in areas with warmer climate" has not been scientifically proven, how about other environmental factors such as pollutions, poor diet, alcohol and drug abuse and so on?

Author Response
Comment: The "low-cost but nevertheless comprehensive methodology" has not been described and shown. Materials and Methods section is brief.
Response: We now specify that the methodology is “capable of evaluating the entire spectrum medically relevant alterations (hot-spot and rare mutations in KRAS, NRAS and BRAF oncogenes; HER2 amplification/overexpression; MSI)”. We have now significantly extended the description of methods. All necessary detail for reproducibility of this methodology are given in Supplementary Table S7. The details of nucleic acids extraction and reverse transcription are now also included in the Materials and Methods.
Comment: The conclusion about "BRAF mutation frequency is a subject to geographical variations, with a relatively low occurrence in areas with warmer climate" has not been scientifically proven, how about other environmental factors such as pollutions, poor diet, alcohol and drug abuse and so on?
Response: As already noticed in our response to the first comment of the Reviewer #1, we are now addressing this issue in the Discussion section:
…Previous studies revealed race-specific differences in the distribution of BRAF mutations, i.e., an increased prevalence of this event in CRC patients of European vs. African or Asian descent [18,23]. Some investigations reported ethnic variations with regard to the frequency of BRAF alterations in CRC observed within the subjects of the same race [24]. In our study, low rate of BRAF oncogene involvement was detected in patients from the North Caucasus and Southern Russia (Figure 4). While the population of the North Caucasus is represented by non-Slavic people of the White race, the ethnic composition of the Southern Russia is identical to other regions of this country. There is also some evidence for a moderate contribution of smoking or dietary factors in determining the probability of development of BRAF-mutated CRC [25-27].
Round 2
Reviewer 1 Report
The authors satisfactorily responded to all requests of the review
Reviewer 2 Report
Discussion and Materials and Methods were significantly improved from the previous version. Authors addressed all concerns of the reviewer.